# Ultrasound-Guided Histotripsy Triggers the Release of Tumor-Associated Antigens from Breast Cancers

**DOI:** 10.3390/cancers17020183

**Published:** 2025-01-08

**Authors:** Shengzhuang Tang, Reliza McGinnis, Zhengyi Cao, James R. Baker, Jr., Zhen Xu, Suhe Wang

**Affiliations:** 1Department of Internal Medicine, Michigan Nanotechnology Institute for Medicine and Biological Sciences, University of Michigan, Ann Arbor, MI 48109, USA; shengzhu@umich.edu (S.T.);; 2Department of Biomedical Engineering, University of Michigan, Ann Arbor, MI 48109, USA; remcginn@umich.edu (R.M.); zhenx@umich.edu (Z.X.)

**Keywords:** histotripsy, tumor antigen, HER2, breast cancer, immune response

## Abstract

Histotripsy is a novel non-invasive ultrasound therapy that mechanically breaks down the target tissue into acellular debris. Histotripsy has been shown to stimulate potent immune response and results in an abscopal effect in rodent tumor models and human case reports. It is hypothesized that the primary mechanism underlying histotripsy immunostimulation is the increased release of tumor antigens by the mechanical disruption of the tumor cell membrane without heating or ionization. This is the first study to quantitatively characterize tumor antigen release and show how it is changed by varying histotripsy dose delivery. Mouse mammary breast tumors were treated with histotripsy, and the extracellular HER2 protein, a key breast tumor antigen, was measured. Regarding extracellular HER2 proteins released from in vitro tumor cells, ex vivo or in vivo tumors by histotripsy had significantly higher levels than untreated cases and depended on the histotripsy dose. These findings provide the first quantitative data on the impact of histotripsy on tumor-associated antigen release and provide important insights into histotripsy immunostimulation.

## 1. Introduction

Conventional treatments for cancer include chemotherapy, radiation therapy, and surgery [1]. These approaches have various drawbacks; however, they provide opportunities to use newer techniques that provide improved efficacy and lower toxicity for cancer therapy. Recent options for cancer treatments include targeted drug therapy, immunotherapy, laser treatments and hormonal therapy, with immunotherapy reflecting a major evolution in our approach to cancer treatment [2,3,4]. However, resistance to cancer immunotherapy frequently occurs due to several different factors including T-cell dysfunction, increased immune regulatory cells, a lack of cellular target expression, deficient cytokine signaling and the formation of a suppressive tumor microenvironment (TME) [4,5]. Importantly, insufficient antigen immunogenicity, the loss of tumor antigens or an inability to present tumor antigens to the immune system have all been described [4,5]. Therefore, new approaches are needed to enhance the efficacy of antigen recognition in cancer immunotherapy.

Histotripsy is a non-invasive ultrasound ablation technology which uses short ultrasound pulses delivered from outside the body and focused into a tumor to generate cavitation and lyse the target into acellular debris [6,7]. It is non-ionizing and does not generate thermal activity. It has demonstrated effectiveness and safety, being approved by the FDA for the clinical treatment of liver tumors [8]. Multiple pre-clinical studies have documented that histotripsy therapy of tumors leads to the release of damage-associated molecular patterns (DAMPs) and inflammatory immune infiltration, which promotes the formation of cytotoxic tumor-infiltrating lymphocytes (TILs) and generally stimulates T-cell immune response [9,10]. The theory behind these observations is that mechanical ablation results in tumor antigen release into the TME that becomes available for antigen-presenting cells (APCs) to present to the adaptive immune system. A recent study reported that tumor antigen peptides released into the TME accumulate in lymphatics, allowing APC antigen uptake and MHC class II presentation [11]. The generated anti-tumor immune response is not locally limited but can treat distal tumors of similar phenotypes, often referred to as an abscopal response [6,9,10,12,13]. Evidence of the abscopal response has been observed in human patients who had liver tumors treated with histotripsy, suggesting the potential for histotripsy to be combined with immunotherapy [11,13].

This study is the first to quantitatively characterize tumor antigen release by histotripsy and show how it is impacted by histotripsy dose delivery. This was studied using the E0771E2 breast cancer cell line with human epidermal growth factor receptor 2 (HER2) as the tumor antigen. HER2 is a transmembrane tyrosine kinase receptor and oncogene that is genetically amplified and overexpressed in about 20% of breast cancers. When activated, it provides the cell with proliferative and anti-apoptosis signals that are a major driver for tumor development and progression [14]. E0771E2 cells were created by engineering mouse mammary tumor E0771 cells with human wildtype HER2. Moreover, tumors (E0771E2 tumor) with stable and functional HER2 expression were formed after inoculating C57BL/6 HER2 transgenic mice with E0771E2 cells [15,16]. We hypothesized that the histotripsy treatment of E0771E2 cells/tumors would result in the release of detectable, cell-free HER2 protein. We therefore performed histotripsy on cultured E0771E2 cells and E0771E2 experimental tumors in mice at varying pulse repetition dosages and analyzed released proteins with Western blot and ultra-performance liquid chromatography (UPLC). Using the recombinant human HER2 extracellular domain (ECD) protein as a reference, this study confirms a dose–response-dependent release of HER2 ECD protein (referred to simply as HER2 or rHER2 for the remainder of this manuscript), which could induce anti-tumor immune responses.

## 2. Materials and Methods

### 2.1. E0771E2 Cell Culture and Animal Studies

E0771E2 cells (frozen at liquid nitrogen) were thawed and then grown in DMEM medium (Thermo Fisher Scientific, Waltham, MA, USA) with 10% FBS (HyClone, GE Healthcare, Chicago, IL, USA) at 37 °C in 5% CO_2_. The cells were harvested and genetically selected by culturing the cells with the drug Geneticin (G418) (Thermo Fisher Scientific, Waltham, MA, USA) at a final concentration of 0.8 mg/mL in DMEM medium with 10% FBS. Six-week-old C57BL/6 HER2 transgenic mice (Jackson Laboratory, Bar Harbor, ME, USA) were inoculated twice with intravenous injections of 10 million of genetically selected HER2-overexpressing E0771E2 cells suspended in 100 µL of phosphate-buffered saline (PBS). The tumor growth was monitored two times per week by measuring the size of the tumor [17].

### 2.2. Flow Cytometry

The level of human HER2 protein expression on EO771E2 cells was analyzed by flow cytometry before inoculation and histotripsy. E0771E2 cells were harvested after three to four cycles of G418 treatments, subsequently washed with 0.1% bovine serum albumin (BSA) in PBS and treated with 2 μg/mL primary mAbs (Trastuzumab, Genentech, Inc, San Francisco, CA, USA) for 30 min at 4 °C. The cells were treated with PE-conjugated mouse anti-human Ig kappa light chain (BD Biosciences, Franklin Lakes, NJ, USA) (1:60). The fluorescence intensities were analyzed using a flow cytometer (Novocyte, Agilent, Santa Clara, CA, USA).

### 2.3. E0771E2 Cell-Culture Sample Preparation

Then, 2 × 10^7^ of E0771E2 cells were suspended in a 5 mL tube with 0.5 mL of 0.1% protease inhibitor (Roche Diagnostics, Indianapolis, IN, USA) in PBS. The cells were then treated with histotripsy pulses with either a low, moderate or high dose (defined in Table 1). The suspensions resulting from the lysis were centrifugated at 12.6 K rpm for 5 min at 4 °C and the supernatant fluids were removed as the cell-free fragment for analysis. The centrifuge sediments were subsequently treated with radioimmunoprecipitation assay buffer (RIPA buffer) (Sigma-Aldrich, St Louis, MO, USA), and then centrifugated at 12.6 K rpm for 5 min at 4 °C. The resulting supernatant fluids were obtained as the cell RIPA lysis for further assay.

### 2.4. Ex Vivo E0771E2 Tumor Sample Preparation

Transgenic HER2 mice were euthanized, and their tumor tissues were resected, suspended in 250 µL of PBS with 0.1% EDTA-free protease inhibitor, and subjected to high doses of histotripsy treatment (Table 1). The tumor tissues were manually dissected and chopped into small pieces. Then, 250 µL of PBS with 0.1% EDTA-free protease inhibitor was added. The samples were centrifugated at 12.6 K rpm for 5 min at 4 °C to yield the supernatants for analysis. Meanwhile, the precipitate sediments were lysed with 250 µL of RIPA buffer, and then centrifugated at 12.6 K rpm for 5 min at 4 °C to yield the supernatants for analysis.

### 2.5. In Vivo E0771E2 Tumor Sample Preparation

Histotripsy-treated mice were euthanized, and their tumor tissues were surgically resected, dissected, and chopped into small pieces. Then, 250 µL of PBS with 0.1% EDTA-free protease inhibitor was added. The samples were centrifugated at 12.6 K rpm for 5 min at 4 °C to yield the supernatants for analysis. Meanwhile, the precipitate sediments were lysed with 250 µL of RIPA buffer, and then centrifugated at 12.6 K rpm for 5 min at 4 °C to yield the supernatants for analysis.

### 2.6. Histotripsy Treatments

In vitro and ex vivo: Histotripsy treatment was performed on all samples using a custom built, 8-element 1 MHz array transducer using 1–2 cycle pulses at 100 Hz pulse repetition frequency (PRF) and an estimated peak negative pressure of ~53 MPa (Figure 1). Samples were targeted via ultrasound imaging using an L40-8/12 20 MHz ultrasound probe (Ultrasonix, Vancouver, Canada) coaxially aligned with the therapy transducer. The treatment pattern was generated by setting the three-dimensional boundaries and populating an ellipsoid grid with equal spacing between points, which was 1 mm for the in vitro treatment and 0.5 mm for the ex vivo tumor treatment. Cell culture samples were gently vortexed to ensure homogenous suspension prior to treatment, while ex vivo tumors were targeted with a sub-total volume to minimize suspended tissue movement during treatment. Three histotripsy doses were delivered by varying the number of pulses per location (10, 50, 100) (Table 1).

In vivo: Mice were anesthetized using 2–5% Fluriso isoflurane (Las Vegas, NV, USA) and positioned for ultrasound-guided histotripsy following the method similar to the in vitro/ex vivo treatments; however, only a single therapeutic dose was tested, and prescribed therapy volumes varied between mice (Table 1). The power supply voltage was lowered compared to the in vitro/ex vivo experiments to avoid off-target cavitation, with an estimated peak negative pressure of ~46 MPa.

### 2.7. Ultra-Performance Liquid Chromatography (UPLC) Assay

UPLC was employed to detect the release of HER2 protein from targeting tumor cells after histotripsy. UPLC analysis was performed in an Acquity System equipped with a C4 BEH column (100 × 2.1 mm, 300 Å) and a photodiode array detector (detection at 285 nm) (Water, Milford, MA, USA). Each sample solution was mixed with a final 20% acetonitrile concentration (6 µL) and injected and eluted at a flow rate of 0.2 mL/min in a linear gradient mode, as reported previously [9,16]. This linear gradient consisted of two mobile solvents, eluent A and B, with each based on water or acetonitrile that contained TFA (0.1% *v*/*v*). The sample elution began with a mobile phase 1% B (0–2.0 min) which was followed by a linear increase to 80% B (13.4 min), a decrease to 50% B (13.8 min), a decrease to 1% B (14.4 min), and finally, an isocratic elution at 1% B (18 min).

### 2.8. Western Blot Assays

Western blot analysis was performed on both cell-free fragments and RIPA lyse supernatants after E0771E2 cell or tumor tissue histotripsy. The protein concentrations were determined by the Pierce BCA assay (Thermo Fisher Scientific, Waltham, MA, USA). Samples with equal amounts of protein (60 µg) in SDS-sample buffer were loaded and run on NuPAGE 10% Bis-Tris gel (Thermo Fisher Scientific, Waltham, MA, USA) and then transferred onto an Immobilon PVDF membrane (Merck Millipore Ltd., Darmstadt, Germany). The membranes were blocked with 5% milk in TBST (Tris-buffered saline containing 0.1% Tween 20) for 1 h at room temperature, stained with primary antibodies (anti-hErbB2/HER2 affinity purified goat IgG) (R&D systems, Minneapolis, MN, USA) overnight at 4 °C, washed 3 times with TBST, stained with secondary antibodies (HRP-conjugated anti-goat IgG, R&D systems, Minneapolis, MN, USA) for 1.5 h at room temperature and detected using chemiluminescence (Amersham imager 600, GE Healthcare Bio-Sciences Corp., Marlborough, MA, USA).

### 2.9. Statistics and Western Blot Analysis

The protein bands were analyzed using GelAnalyzer 23.1.1 (available at www.gelanalyzer.com) by Istvan Lazar Jr., PhD and Istvan Lazar Sr., PhD, CSc and following the provided user guide. Western blot images were cropped to the region of interest and automatic band detection was used. Bands along the same horizontal axis were made equal width with the HER2 standard lane being used for reference. Baseline subtraction was implemented for each band using default parameters (rolling ball method with peak width tolerance of 39%) and the resulting curve volume calculated. Fold change in the individual band(s) volume was calculated by dividing by the volume of the horizontal matching band in the untreated control (UH) lane as the baseline.

## 3. Results

### 3.1. HER2 Expression in EO771E2 Cells

The E0771E2 cells for histotripsy application were treated with G418 to enhance HER2^+^ levels detected by flow cytometry, which did result in an increase in HER2 protein expression (Figure 2). However, treatment with even higher G418 treatment concentrations (>2.0 mg/mL) led to non-specific high levels of cytotoxicity and fewer cells harvested. Therefore, E0771E2 cells treated with four cycles of 0.8 mg/mL G418 were used for histotripsy treatment or to induce HER2^+^ tumors.

#### 3.1.1. Protein Concentrations in Total Cell-Free Fragments (CFs) and RIPA Buffer Lysate (RB)

The Pierce BCA Protein Assay Kit was used to determine the total protein concentration after the histotripsy treatment of E0771E2 cells (Figure 3). While non-histotripsy cells displayed very lowed protein levels, all histotripsy treatments released significantly more proteins in their supernatants compared to the non-histotripsy cells. High-dose histotripsy triggered significantly more cell lysis and released more proteins into the supernatants compared to low-dose histotripsy (Figure 3A). However, the total released proteins did not linearly increase with histotripsy dose. As expected, an inverse relationship was seen for the total remaining protein in the RIPA lysis buffer-treated cell pellets, with a significant (*p* < 0.05, *T*-test) decrease in total protein in high-dose and moderate-dose histotripsy groups compared to low-dose histotripsy and non-histotripsy groups (Figure 3B).

#### 3.1.2. UPLC Detection Reveals Dose–Response Release of HER2 Protein

UPLC analysis of the E0771E2 lysate displayed a heterogeneous rHER2 protein composition after histotripsy treatment. Both cell-free and RIPA lysis buffer exhibited a peak of free rHER2 protein at ~9.3 min (Figure 4), indicating that the released rHER2 protein had the same size and structure. Moreover, histotripsy treatment led to a dose-dependent triggered release of rHER2 in the cell-free fragment (Figure 4A) and cell RIPA lysis lysate components (Figure 4B), with the higher dose of histotripsy resulting in a higher release of rHER2. It was noted that the levels of released rHER2 protein appeared to be higher (~1–3×-fold increase) in the cell-free fragment buffer (Figure 4A) than in the cell RIPA lysis buffer (Figure 4B).

#### 3.1.3. Western Blot Assay of the Dose–Response Release of HER2 In Vitro Histotripsy of E0771E2 Cells

A Western blot assay was used to confirm the presence of minimally degraded rHER2 peptide in the generated lysates with purified rHER2 peptide used as a positive control and vinculin as a loading control (Figure 5B and Figure 6B). Using the monoclonal antibody against rHER2 protein, a band appeared at 110 kDa (Figure 5A, Lane 2). A protein band of the same size was found in the cell-free fragments (CFs) of E0771E2 cells treated with histotripsy (Lanes 4–6, Figure 5A) while this specific band was barely detected in the untreated cells (Lanes 3 and 7), documenting the histotripsy-mediated release of HER2 from cells. Additionally, the 110 kDa protein band intensity was dose-dependent, with a 0.5×, ~3.5×, and ~5×-fold increase in density for the low (Lane 4), moderate (Lane 5), and high (Lane 6) histotripsy doses, respectively, compared to the untreated sample (Lane 3). A matching 110 kDa band was also detected in the RIPA lysis buffer component of E0771E2 cells treated with histotripsy, with the band density increasing by 1.4×, 2.2×, and 3× in the low, moderate, and high histotripsy dose groups, respectively, compared to the band in the untreated sample (Figure 6A, Lanes 4-6).

### 3.2. Ex Vivo Histotripsy of E0771E2 Tumors

Having obtained evidence of HER2 peptide being released from cultured E0771E2 cells treated with histotripsy, we next sought to determine if the histotripsy treatments of ex vivo E0771E2 tumors resected from HER2 transgenic mice resulted in liberated rHER2 in the extracellular space. Like the in vitro experiments, total peptide and HER2 presence were analyzed using BCA, UPLC, and Western blotting assays.

#### 3.2.1. BCA Protein Assay of Tumor Histotripsy Treatment Lysis

A BCA protein assay was used to determine the total protein concentration in E0771E2 tumors treated with histotripsy (Figure 7). We observed a dose-dependent release of total protein, with more tumor protein detected in cell-free supernatants (Figure 7B) than in RIPA lysis buffer of the cell pellets (Figure 7C). There was no significant difference in total protein in the cell-free component between the low-dose and untreated group; however, there was a significant increase in the moderate and high treatment groups, with a ~1.5×-fold increase in total protein compared to the untreated sample. In the RIPA lysis buffer lysate component, we report an inverse relationship in total protein concentration compared to the cell-free samples, with a statistically significant dose-dependent decrease in peptide concentration as histotripsy dose increased.

#### 3.2.2. UPLC Analysis of the Lysis from the Tumor Treated with Histotripsy

Again, UPLC analysis showed that the histotripsy-treated ex vivo tumor cell-free supernatants and RIPA lysis buffer lysate components exhibited a peak of free rHER2 protein at ~9.3 min (Figure 8). A histotripsy dose–response release of HER2 was also evident in both the cell-free fragment buffer (Figure 8A) and cell RIPA lysis buffer (Figure 8B). However, compared with the levels of released HER2 protein in the cell RIPA lysis buffer, a higher level of the released HER2 protein was found in the cell-free fragment buffer only at the higher histotripsy dose (Figure 8).

#### 3.2.3. Western Blot Assay of Tumor Histotripsy Treatment Lysis

rHER2 protein (110 kDa) was detected in the cell-free (CF) component of E0771E2 tumors treated with histotripsy (Figure 9A, Lanes 4–6). This analysis showed a ~5.4×, ~6.8×, and ~28× increase in band density for the LH, MH, and HH doses, respectively, compared to the untreated control. The 110 kDa rHER2 band was consistently weaker in the RB component of treated tumors (Figure 10A, Lanes 4–6), which showed no difference in band density in the LH (Lane 4) or MH (Lane 5) doses compared to the untreated sample (Lane 3). There was also only a ~2.8× increase in band density in the HH dose (Lane 6) compared to the untreated sample. Vinculin band intensity corroborates equal peptide loading across wells (Figure 9B and Figure 10B).

### 3.3. In Vivo Histotripsy of E0771E2 Tumors

To demonstrate the feasibility of probing rHER2 in vivo, HER2 tumor-bearing mice were either treated with the highest histotripsy dose from the previous experiments (HT) or untreated controls (UH) (Figure 11). The overall protein concentration of the cell-free lysate increased ~3× compared to the untreated control (Figure 11C), and a similar elution spike was seen at ~9.3 min compared to the rHER2 standard (Figure 11B). Interestingly, the treated in vivo cell-free lysate did not show any observable band at the same molecular weight as the rHER2 standard (110 KDa); however, bands at the ~50–60 KDa MW had a ~42× increase in relative density compared to untreated bands (Figure 11D).

## 4. Discussion

The goal of this study was to quantitatively characterize the ability of histotripsy to release a putative cancer antigen (HER2) from tumor cells. In addition, we wanted to quantitatively probe for released tumor-specific antigens (HER2 extracellular domain) in the extracellular supernatants and tissue following in vitro and in vivo histotripsy treatment as a function of dose. We were able to document, for the first time, that histotripsy treatment significantly increases the release of total antigenic HER2 into the extracellular compartment in a dose-dependent manner, as confirmed by the protein assay, UPLC analysis, and Western blot. This is the first study to quantitatively demonstrate a dose-dependent release of a tumor-associated antigen (HER2) following histotripsy therapy of tumor cells, suggesting that histotripsy could release tumor antigens in a dose-dependent manner.

Preclinical and clinical studies have indicated that histotripsy is able to stimulate local and systemic immune responses, leading to the reduction in off-target tumors and metastases (abscopal effects) [6,9,11,12,13]. One major mechanism thought to underly histotripsy-induced abscopal effects is the mechanical disruption of tumor tissues and cells that release “biologically available” tumor-specific antigens (TSAs) or tumor-associated antigens (TAAs). The release of intact TSAs or TAAs differentiates histotripsy from thermal techniques and radiation therapy, which usually denature these antigens and disrupt both antibody and cellular epitopes. This was reinforced by a recent report documenting intact ZsG antigens being released from tumors treated with histotripsy ablation accumulating and being processed by APCs in the tumor draining lymph node [11]. In this study, we quantify HER2 release from positive mammary tumor cells to demonstrate that histotripsy treatment could trigger the release of HER2 in vitro in cultured cells as well as ex vivo in a murine tumor model in a dose-dependent manner. In addition, the released HER2 appears to be antigenically intact as it is the same molecular weight as the rHER2 protein standard, and both were detected by HER2 mono-antibody and UPLC.

HER2 is one of the most studied TAAs as a cancer target and for immunotherapies [18,19]. HER2 is a valid TAA therapeutic target based on at least two concepts. First, HER2 overexpression is observed in a significant number of human cancers including breast cancer, and importantly, its overexpression plays a critical role in the development and survival of cancers. Second, the expression of HER2 is relatively lower in normal tissues. Given these issues, various approaches have been developed to inhibit HER2 expression or inactivate its activity, which leads to the arrest of tumor growth [14,15,18,19,20]. Currently, there are no reports on the use of histotripsy to release HER2. Three methods have been employed in this study to analyze the histotripsy-mediated release of HER2 protein from cultured tumor cells and ex vivo or in vivo tumors. We definitively showed HER2 release from the cell pellet into the extracellular compartment in vitro, ex vivo, and in vivo.

Many tumor antigens are only weakly immunogenic and fail to induce anti-tumor immune response in vivo. This is thought to be the case for HER2, a poorly immunogenic TAA which can induce a weak or undetectable immune response in human and animal studies [21,22]. There are several factors that may contribute to poor immunogenicity, including low concentrations of antigens and an immunosuppressive tumor microenvironment (TME) and lack of interaction with immune cells [23,24,25]. The histotripsy-mediated release of antigenically intact HER2 ECD from the cell membrane into the extracellular compartment could facilitate the presentation of HER2 antigens to the immune system. Combined with the potential disruption of the immunosuppressive TME seen in EO771 tumors (and other breast cancer types), antigen release may contribute to the induction of an abscopal effect and enhanced anti-tumor immune response with histotripsy treatment [26,27,28]. HER2 was the only TAA probed in this study due to its overexpression in our cell line and an already existing HER2 workflow for Western blot and HPLC detection. However, we believe the dose-dependent relationship related to tumor antigen release by histotripsy may also be expanded to other tumor types and antigen targets. In fact, the association between the abscopal effect and the release of TAAs has been documented in tumors treated by other anti-tumor methods such as photothermal therapy [29,30,31].

To verify histotripsy-mediated HER2 release in vivo, we established a mouse tumor model and treated it with histotripsy. When these tumors were subjected to sonic disruption treatment in vivo, there was a significant increase in HER2 protein in the cell-free fragments from recovered tumors compared to untreated controls. This supports the concept of tumor-associated antigen (TAA) release in vivo following histotripsy treatment. Also, a consistent and robust dose of energy could be applied, with the tumor proximal to the sonic generator during treatment. In future, more complex in vivo studies will be carried out, and HER2 release will be comprehensively investigated across histotripsy doses and tumor sizes. We will also investigate the immunogenic potential of histotripsy-generated lysates, focusing on their ability in vivo to stimulate antigen-presenting cells and cognate B cells, and we will characterize the immune response to the released HER2 protein.

## 5. Conclusions

In summary, we have quantitatively and qualitatively demonstrated that histotripsy treatment triggers HER2 release from tumor cells into the extracellular compartment. The histotripsy-mediated the release of HER2 antigens could be therapeutically relevant in the treatment of HER2^+^ cancers including breast cancer, gastroesophageal adenocarcinomas, and colon carcinoma. Importantly, the dose-dependent release of tumor-associated antigens may apply generally to the histotripsy treatment of many tumor types and provide important insights into the mechanism underlying histotripsy immune stimulation.

## Figures and Tables

**Figure 1 cancers-17-00183-f001:**
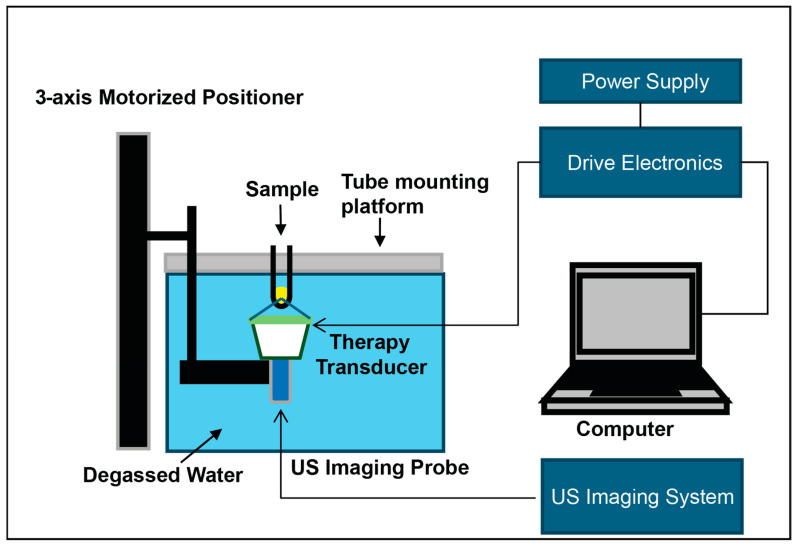
Diagram of histotripsy setup. Samples are positioned above a degassed water tank with the histotripsy transducer and ultrasound imaging probe angled upwards towards the sample.

**Figure 2 cancers-17-00183-f002:**
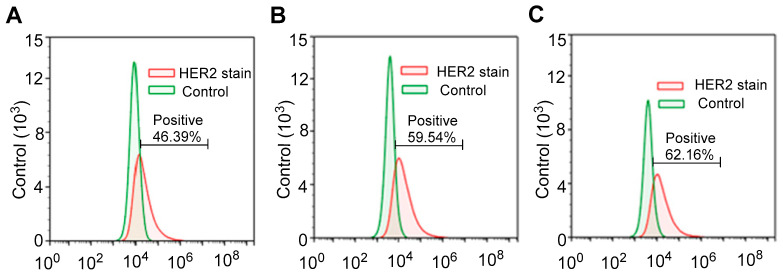
(**A**–**C**) Flow cytometry evaluation of HER2^+^ expression in growing E0771E2 cells treated with various G418 concentrations: 0.5, 0.8, and 1.2 mg/mL.

**Figure 3 cancers-17-00183-f003:**
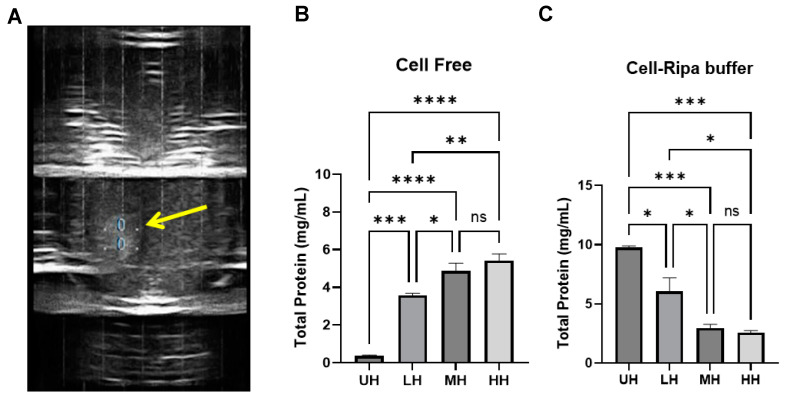
(**A**) A representative ultrasound image of the bubble cloud in the sample tube during treatment (yellow arrow points to bubble cloud). Total protein levels of histotripsy treatment cells in (**B**) cell-free fragments, and (**C**) RIPA lysis buffer quantified by BCA Protein Assay Kit. * *p* < 0.05, ** *p* < 0.01, *** *p* < 0.001, **** *p* < 0.0001, ns: no significance, *t*-test.

**Figure 4 cancers-17-00183-f004:**
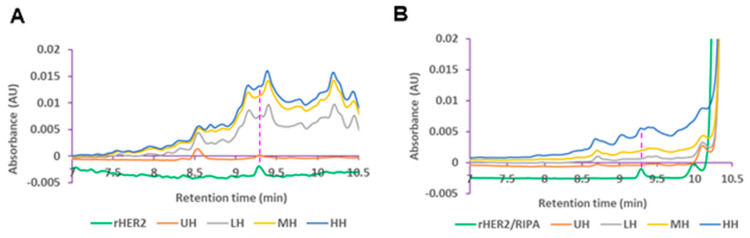
Released HER2 levels of histotripsy-treated cells in cell-free fragments (**A**) and RIPA lysis buffer (**B**) detected by ultra-performance liquid chromatography (UPLC) at 285 nm.

**Figure 5 cancers-17-00183-f005:**
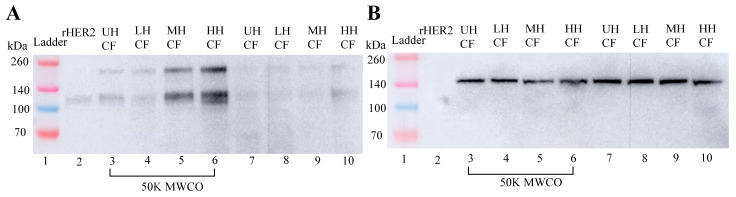
(**A**) Western blot analysis of histotripsy cell-free fragment (CF) probed with monoclonal HER2 antibody (**B**) with a vinculin antibody blot documenting the protein loading. Lanes 1 and 2 were ladder (13 µL) and rHER2 (0.01 µg), respectively, as references. Lane 3 was an untreated control. Lanes 4–10 analyzed the cell-free fragments of E077E2 cells with a 20 µg/lane protein loaded. Lanes 7–10 directly analyzed cell-free lysate from untreated (UH) and low (LH), moderate (MH) and high (HH) dose of histotripsy treatments, respectively; lanes 3–6 were the corresponding lysate after 50 kDa molecular weight cut-off membrane dialysis at 4 °C with 0.1% protease inhibitor in PBS as the dialysate. The uncropped blots are shown in the Appendix A.

**Figure 6 cancers-17-00183-f006:**
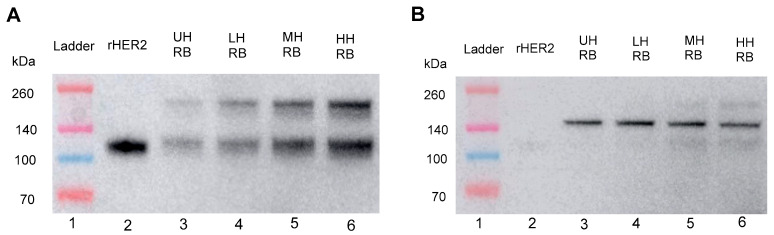
Western blot analysis of histotripsy cell RIPA lysis buffer probed with (**A**) monoclonal HER2 antibody and (**B**) vinculin, documenting the loading control. Lanes 1 and 2 were ladder (13 µL) and rHER2 (0.012 µg), respectively, as references. Lanes 3–6 were the analysis for cell lysis with 15 µg of protein per lane. Lane 3 was untreated, and lanes 4–6 were the low-, moderate-, and high-dose histotripsy treatment, respectively. The uncropped blots are shown in the Appendix A.

**Figure 7 cancers-17-00183-f007:**
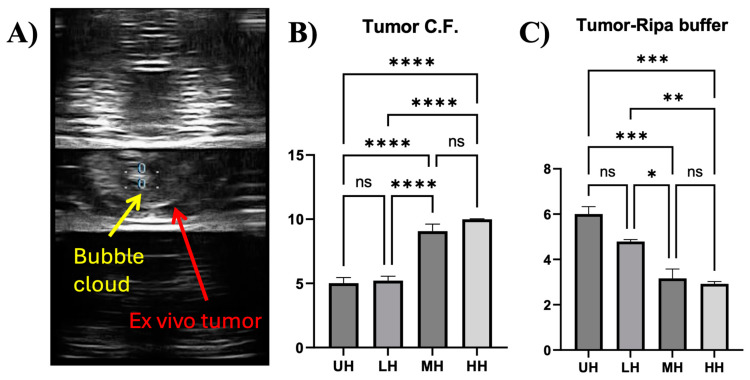
(**A**) Representative ultrasound image of the histotripsy-generated bubble cloud (yellow arrow) in the ex vivo tumor sample (red arrow) during treatment. Total protein levels of histotripsy treatment cells in (**B**) tumor free fragments, and (**C**) RIPA lysis buffer quantified by BCA Protein Assay Kit. *n* = 6. * *p* < 0.05, ** *p* < 0.01, *** *p* < 0.001, **** *p* < 0.0001, ns: no significance, *t*-test.

**Figure 8 cancers-17-00183-f008:**
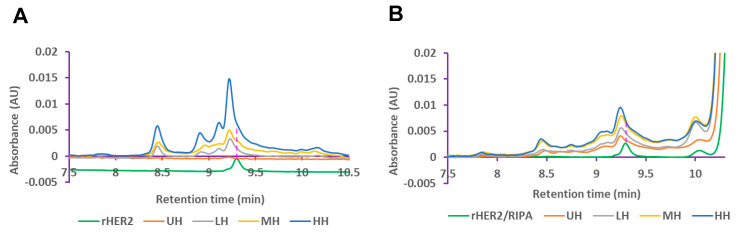
Released HER2 levels of histotripsy-treated tumors in cell-free fragments (**A**), and RIPA lysis buffer (**B**) detected by UPLC at 285 nm.

**Figure 9 cancers-17-00183-f009:**
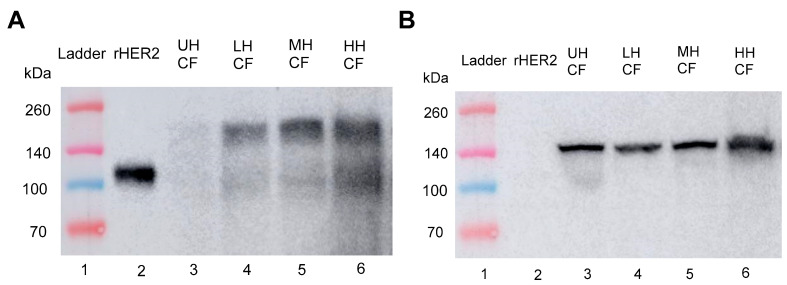
Western blot detection of ex vivo histotripsy E0771E2 tumor cell-free fragments/supernatants probed with (**A**) monoclonal HER2 antibody, and (**B**) the loading control vinculin antibody. Lanes 1 and 2 were ladder (13 µL) and positive control rHER2 (0.015 µg), respectively, as references. Lanes 3–6 were the cell-free fragments of E077E2 tumors with 30 µg/lane protein loading for un-treatment (UH) and a low (LH), moderate (MH), and high (HH) dose of histotripsy treatment, respectively. The uncropped blots are shown in the Appendix A.

**Figure 10 cancers-17-00183-f010:**
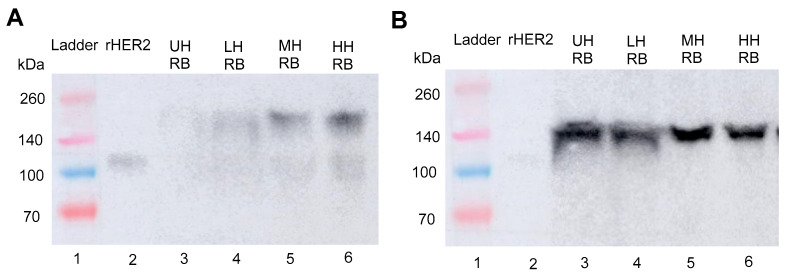
Western blot detection of ex vivo histotripsy E0771E2 tumor RIPA lysis buffer probed with (**A**) monoclonal HER2 antibody and (**B**) the loading control vinculin. Lanes 1 and 2 were ladder (13 µL) and the positive control rHER2 (0.01 µg), respectively, as references. Lanes 3–6 were the RIPA lysis buffer for E077E2 tumors with 30 µg/lane protein loading for untreated (UH) and a low (LH), moderate (MH), and high (HH) dose of histotripsy treatment, respectively. The uncropped blots are shown in the Appendix A.

**Figure 11 cancers-17-00183-f011:**
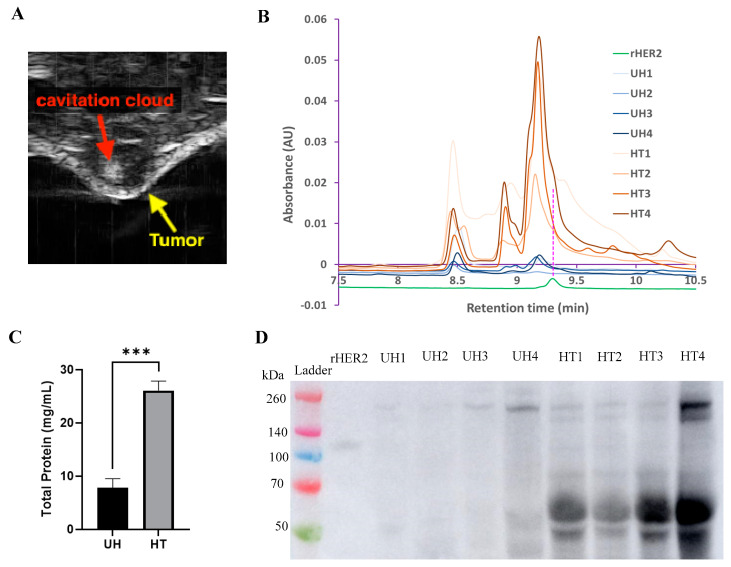
HER2 protein detected after in vivo histotripsy treatment. (**A**) Representative ultrasound image of the histotripsy cavitation cloud, in vivo, during treatment. Analysis of in vivo histotripsy E0771E2 tumor cell-free fragments/supernatants with (**B**) UPLC assay, detected at 285 nm; (**C**) BCA protein assay. (**D**) Western blot assay probed with monoclonal HER2 antibody. Ladder was loaded with 15 µL/lane; positive control rHER2 was loaded with 0.01 µg/lane; samples were loaded with 50 µg/lane protein. *n* = 4. *** *p* < 0.001, *T*-test. The uncropped blots are shown in the Appendix A.

**Table 1 cancers-17-00183-t001:** Summary of histotripsy dose conditions.

Experiment	Pulses per Location	3D Location Spacing (mm)	Target Volume (mm^3^)	Total Locations	Total Pulses	Pulse Density (Pulses/mm^3^)
In vitro	10 (LH) 50 (MH) 100 (HH)	1	100.5	99	990 (LH) 4950 (MH) 9900 (HH)	10 (LH) 49 (MH) 98 (HH)
Ex vivo		0.5	25	191	1910 (LH) 9550 (MH) 19,100 (HH)	76 (LH) 380 (MH) 760 (HH)
In vivo	100	0.5	17–52.5	125–417	12,500–41,700	863–796

LH: low-dose histotripsy; MH: moderate-dose histotripsy; HH: high-dose histotripsy.

## Data Availability

Data are contained within the article.

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
