# Peer review of "Ultrasound-Guided Histotripsy Triggers the Release of Tumor-Associated Antigens from Breast Cancers"

_cancers, 2025, doi:10.3390/cancers17020183_

Round 1
Reviewer 1 Report (Previous Reviewer 1)
Comments and Suggestions for Authors
In the present study, authors demonstrated the ability of histotripsy to release a putative cancer antigen (HER2) from tumor cells. Authors were able to document that histotripsy treatment significantly increases the release of total antigenic HER2 into the extracellular compartment in a dose-dependent manner, as confirmed by the protein assay, UPLC analysis, and Western blot.
Im my opinion, the present manuscript is well written and data are clearly presented.
It can be suitable for publication in its present form.
Author Response
We thank Reviewer 1 for his/her careful assessment and professional comments.
Reviewer 2 Report (Previous Reviewer 2)
Comments and Suggestions for Authors
in this reversion - the authors address some minor problems; but the design of this studies limited its application and translational value.
Author Response
Reviewer 2
Comments: in this reversion - the authors address some minor problems; but the design of this studies limited its application and translational value.
Response: Though the reviewer did not specifically explain the problem of the study design this time, by checking the previous comments, we believe, the reviewer is still concerned abouts HER2 as a therapeutic target as he/she indicated in the 1st round comments “HER2 is expressed in normal epithelial cells across several organs, including the gastrointestinal, respiratory, reproductive, and urinary tracts. Targeting HER2 has been well-contested in the literature due to the potential for severe toxicity stemming from off-target effects on normal tissues. Consequently, the project's premise is flawed, even in the best-case scenario. Even if histotripsy were to increase HER2 expression in vivo (which the authors have not demonstrated in this paper), it is unlikely to translate into a viable therapeutic strategy due to these inherent risks.”. We addressed this issue in detail in our previous response and we would like to have our further response below:
It is true that HER2 as a TAA can be expressed in normal tissues, but its level is lower than in tumor tissues, and that targeting HER2 may induce side-effects or off-target effects. However, the benefits of anti-HER2 therapies significantly outweigh the known side effects, which leads to the development of increasing anti-HER2 treatments including well-known herceptin/ trastuzumab, margenza, perjeta/pertuzumab, and nerlynx/neratinib. Importantly, all these agents were granted by FDA to treat patients with HER2+ breast cancer in 1998, 2006, 2017 and 2020 respectively. By searching PubMed (Nov 2024): title or abstract containing anti-HER2 or HER2 targeting, there are 35406 publications. These anti-HER2 therapies have significantly improved the prognosis of patients with HER2-positive breast cancer and below are just a few relevant references (1-6).
- DuMond B, Patel V, Gross A, Fung A, Weber S. Fixed-dose combination of pertuzumab and trastuzumab for subcutaneous injection in patients with HER2-positive breast cancer: A multidisciplinary approach. J Oncol Pharm Pract. 2021 Jul;27(5):1214-1221. doi: 10.1177/1078155221999712. Epub 2021 Mar 9. PMID: 33719721.
- Saleh K, Khoury R, Khalife N, Chahine C, Ibrahim R, Tikriti Z, Le Cesne A. Mechanisms of action and resistance to anti-HER2 antibody-drug conjugates in breast cancer. Cancer Drug Resist. 2024 Jun 3;7:22. Doi: 10.20517/cdr.2024.06. PMID: 39050884; PMCID: PMC11267152.
- Tommasi C, Airò G, Pratticò F, Testi I, Corianò M, Pellegrino B, Denaro N, Demurtas L, Dessì M, Murgia S, Mura G, Wekking D, Scartozzi M, Musolino A, Solinas C. Hormone Receptor-Positive/HER2-Positive Breast Cancer: Hormone Therapy and Anti-HER2 Treatment: An Update on Treatment Strategies. J Clin Med. 2024 Mar 24;13(7):1873. doi: 10.3390/jcm13071873. PMID: 38610638; PMCID: PMC11012464.
- Stanowicka-Grada M, Senkus E. Anti-HER2 Drugs for the Treatment of Advanced HER2 Positive Breast Cancer. Curr Treat Options Oncol. 2023 Nov;24(11):1633-1650. doi: 10.1007/s11864-023-01137-5. Epub 2023 Oct 25. PMID: 37878202; PMCID: PMC10643304.
- Wu XM, Qian YK, Chen HL, Hu CH, Chen BW. Efficacy and Safety of Anti-HER2 Targeted Therapy for Metastatic HR-Positive and HER2-Positive Breast Cancer: A Bayesian Network Meta-Analysis. Curr Oncol. 2023 Sep 15;30(9):8444-8463. doi: 10.3390/curroncol30090615. PMID: 37754530; PMCID: PMC10528081.
- Giugliano F, Carnevale Schianca A, Corti C, Ivanova M, Bianco N, Dellapasqua S, Criscitiello C, Fusco N, Curigliano G, Munzone E. Unlocking the Resistance to Anti-HER2 Treatments in Breast Cancer: The Issue of HER2 Spatial Distribution. Cancers (Basel). 2023 Feb 22;15(5):1385. doi: 10.3390/cancers15051385. PMID: 36900178; PMCID: PMC10000152.
Nevertheless, we thank Reviewer 2 for his/her comments but we cannot agree with his/her point. We believe, our point is in line with most of publications as well as FDA.
This manuscript is a resubmission of an earlier submission. The following is a list of the peer review reports and author responses from that submission.
Round 1
Reviewer 1 Report
Comments and Suggestions for Authors
In the present paper authors demonstate that histotripsy treatment significantly increases the release of total antigenic HER2 into the extracellular compartment in a dose-dependent manner, as confirmed by the protein assay, UPLC analysis, and Western blot.
Manuscript is well written and data are clearly presented. I have a few suggestions to improve the overall quality.
1) Scientific evidence suggests that histotripsy is able to stimulate local and systemic immune responses. Authoprs should explain why they limited their search to HER2 and if other TSA were initially considered for evaluation.
2) Tumor microenviroment and PD-L1 expression in BC should be disccussed and correlated to author's findings (please refer to PMID: 37760449)
Author Response
We greatly appreciate the reviewers’ comments. We have addressed all reviewers’ comments as detailed below (response in Bold)
--------------Reviewer 1---------------
In the present paper authors demonstrate that histotripsy treatment significantly increases the release of total antigenic HER2 into the extracellular compartment in a dose-dependent manner, as confirmed by the protein assay, UPLC analysis, and Western blot.
Manuscript is well written and data are clearly presented. I have a few suggestions to improve the overall quality.
1) Scientific evidence suggests that histotripsy is able to stimulate local and systemic immune responses. Authors should explain why they limited their search to HER2 and if other TSA were initially considered for evaluation.
Response: HER2 was used as an example of TAA to prove the concept of correlation between HT doses and TAA release, because it was well established in our lab. Other TSA’s are considered for other tumor models, which we plan to investigate in the future.
2) Tumor microenviroment and PD-L1 expression in BC should be discussed and correlated to author's findings (please refer to PMID: 37760449)
Response: We thank the reviewer for the insightful comment and suggestion. Importance of TILs and PD-L1 expression in breast cancer has been addressed in the discussion section (the 1st Para, Page 14) and PMID: 37760449 has been added as Ref 29 in the list.
Reviewer 2 Report
Comments and Suggestions for Authors
In this manuscript, the authors aim to demonstrate that histotripsy can be used to increase HER2 surface expression, thereby enhancing antigen recognition of tumors and potentially improving immunotherapy outcomes. However, the study suffers from significant theoretical and experimental flaws that limit its contribution to the scientific community.
First, the theoretical foundation of the study is problematic, starting from the claim in line 28 of the abstract: "...HER2, a well-defined TSA target for cancer..." This statement is inaccurate, as HER2 is NOT a tumor-specific antigen. HER2 is expressed in normal epithelial cells across several organs, including the gastrointestinal, respiratory, reproductive, and urinary tracts. Targeting HER2 has been well-contested in the literature due to the potential for severe toxicity stemming from off-target effects on normal tissues. Consequently, the project's premise is flawed, even in the best-case scenario. Even if histotripsy were to increase HER2 expression in vivo (which the authors have not demonstrated in this paper), it is unlikely to translate into a viable therapeutic strategy due to these inherent risks.
The experimental design also raises significant concerns. All of the experiments were conducted in vitro, using a setup that lacks any clear translational relevance to in vivo conditions. The authors have not adequately explained how their findings could be applied in vivo, nor have they addressed whether the same effects would be observed in a living system. There is also no discussion of the technical barriers preventing the authors from conducting in vivo experiments, which would be crucial for assessing the translational potential of their findings.
In summary, the manuscript suffers from fundamental issues in both its theoretical premise and experimental design, significantly limiting its value and impact.
Author Response
--------------Reviewer 2---------------
In this manuscript, the authors aim to demonstrate that histotripsy can be used to increase HER2 surface expression, thereby enhancing antigen recognition of tumors and potentially improving immunotherapy outcomes. However, the study suffers from significant theoretical and experimental flaws that limit its contribution to the scientific community.
First, the theoretical foundation of the study is problematic, starting from the claim in line 28 of the abstract: "...HER2, a well-defined TSA target for cancer..." This statement is inaccurate, as HER2 is NOT a tumor-specific antigen. HER2 is expressed in normal epithelial cells across several organs, including the gastrointestinal, respiratory, reproductive, and urinary tracts. Targeting HER2 has been well-contested in the literature due to the potential for severe toxicity stemming from off-target effects on normal tissues. Consequently, the project's premise is flawed, even in the best-case scenario. Even if histotripsy were to increase HER2 expression in vivo (which the authors have not demonstrated in this paper), it is unlikely to translate into a viable therapeutic strategy due to these inherent risks.
The experimental design also raises significant concerns. All of the experiments were conducted in vitro, using a setup that lacks any clear translational relevance to in vivo conditions. The authors have not adequately explained how their findings could be applied in vivo, nor have they addressed whether the same effects would be observed in a living system. There is also no discussion of the technical barriers preventing the authors from conducting in vivo experiments, which would be crucial for assessing the translational potential of their findings.
In summary, the manuscript suffers from fundamental issues in both its theoretical premise and experimental desig, significantly limiting its value and impact.
Response: Thank you for your clarification of term for tumor specific antigen. We have amended HER2 as a tumor associate antigen (TAA). We did not demonstrate that histotripsy can increase HER2 surface expression, however, we validated release of HER2 from breast tumor cells treated by histotripsy.
It is true that HER2 and other TAAs are also expressed in non-tumor tissues (usually much lower than in tumor tissues) and that targeting these TAAs may induce side-effects. It is well known that there are some significant treatments to target some of these TAAs. For example, trastuzumab (i.v. or subcutaneous injection) is one of the known agents to target HER2 for breast cancer (Ref below 1).
It should be noted that in our case, we did not inhibit HER2 to arrest the tumor but we demonstrated that HER2 (or other TAAs) can be released from tumor cells treated by histotripsy. The HER2 release is likely the potential mechanism contributing to the abscopal effect observed in histotripsy-treated tumors (Ref 11-13). The association between the abscopal effect and the release of TAAs has been well reported in other anti-tumor treatments (Ref below 2-4).
- DuMond B, Patel V, Gross A, Fung A, Weber S. Fixed-dose combination of pertuzumab and trastuzumab for subcutaneous injection in patients with HER2-positive breast cancer: A multidisciplinary approach. J Oncol Pharm Pract. 2021 Jul;27(5):1214-1221. doi: 10.1177/1078155221999712. Epub 2021 Mar 9. PMID: 33719721.
- Franzese O, Torino F, Giannetti E, Cioccoloni G, Aquino A, Faraoni I, Fuggetta MP, De Vecchis L, Giuliani A, Kaina B, Bonmassar E. Abscopal Effect and Drug-Induced Xenogenization: A Strategic Alliance in Cancer Treatment? Int J Mol Sci. 2021 Oct 1;22(19):10672. doi: 10.3390/ijms221910672. PMID: 34639014; PMCID: PMC8509363.
- Xia QH, Lu CT, Tong MQ, Yue M, Chen R, Zhuge DL, Yao Q, Xu HL, Zhao YZ. Ganoderma Lucidum Polysaccharides Enhance the Abscopal Effect of Photothermal Therapy in Hepatoma-Bearing Mice Through Immunomodulatory, Anti-Proliferative, Pro-Apoptotic and Anti-Angiogenic. Front Pharmacol. 2021 Jul 6;12:648708. doi: 10.3389/fphar.2021.648708. PMID: 34295244; PMCID: PMC8290260.
- Chen M, Quan G, Wen T, Yang P, Qin W, Mai H, Sun Y, Lu C, Pan X, Wu C. Cold to Hot: Binary Cooperative Microneedle Array-Amplified Photoimmunotherapy for Eliciting Antitumor Immunity and the Abscopal Effect. ACS Appl Mater Interfaces. 2020 Jul 22;12(29):32259-32269. doi: 10.1021/acsami.0c05090. Epub 2020 May 22. PMID: 32406239.
Some of the above information have been added into the revised manuscript (the last Para, Page 13 and the 1st Para, Page 14).
Reviewer 3 Report
Comments and Suggestions for Authors
In this study, the authors investigated the effects of histotripsy, a novel non-invasive ultrasound therapy, on the release of the tumor-specific antigen HER2 from breast tumor cells. They demonstrated that histotripsy significantly increases the extracellular release of HER2 protein both in vitro and ex vivo, with the amount of HER2 released being dependent on the dose of histotripsy applied. This provides important insights into how histotripsy may enhance immune responses by promoting tumor antigen release, thereby supporting its potential role in cancer immunotherapy. I have few suggestions for the improvement.
1) How stable is HER2 expression in the E0771E2 cells over time, and how consistent is it in vivo in the C57BL/6 HER2 transgenic mice? Did you observe any variation in expression levels, and if so, how did you account for this in your analysis?
2) How were the different pulse repetition dosages of histotripsy selected? Were these doses chosen based on preliminary experiments, and how do they compare to clinically relevant parameters?
3) How does tumor heterogeneity affect the release of HER2 after histotripsy? Were any studies performed to determine if different regions of the tumor had different responses to histotripsy?
4) Is there any direct evidence from this study that histotripsy-induced HER2 release leads to MHC class I or II presentation by APCs?
5) How does histotripsy impact other components of the tumor microenvironment, such as immune suppressive cells or stromal components? Were these effects considered in relation to the release of HER2 and potential immunostimulation?
6) The introduction mentions abscopal effects, but does this study provide direct evidence of these effects in the animal model used? If not, how might future studies be designed to assess these effects systematically?
7) The manuscript suggests that released tumor antigens lead to APC presentation and subsequent T cell activation. Any functional assays (e.g., mixed lymphocyte reaction) to assess whether T cells were activated by these released antigens?
8) What are some of the limitations of using the E0771E2 mouse model for studying HER2-positive cancer in the context of immunotherapy? How might these limitations affect the interpretation of your findings, and what future work do you propose to address these limitations?
9) Were any experiments conducted to determine whether histotripsy selectively targets tumor cells without damaging surrounding normal tissue? This could provide insight into the safety of histotripsy in a clinical context.
10) The study mentions that G418 was used at different concentrations, but higher concentrations caused significant cytotoxicity. Could you elaborate on how the chosen concentration (0.8 mg/mL) balances HER2 enhancement without excessive cell death? How reproducible is this effect across different batches of cells?How might the observed levels of released HER2 affect its immunogenicity in the tumor microenvironment? Was there any attempt to correlate the amount of HER2 released with potential immune activation?
11) The study shows higher HER2 levels in the cell-free fraction compared to RIPA lysate. Could you elaborate on the mechanistic implications of this finding? Does histotripsy preferentially release HER2 into the extracellular environment rather than leaving it within the remaining cell lysate?
12) The Western blot results indicate a substantial increase in HER2 release at the high histotripsy dose (~28x increase). Could high-dose histotripsy also induce non-specific protein release or damage to adjacent non-tumor cells? How might this affect the interpretation of HER2 specificity in the tumor microenvironment?
Author Response
--------------Reviewer 3---------------
In this study, the authors investigated the effects of histotripsy, a novel non-invasive ultrasound therapy, on the release of the tumor-specific antigen HER2 from breast tumor cells. They demonstrated that histotripsy significantly increases the extracellular release of HER2 protein both in vitro and ex vivo, with the amount of HER2 released being dependent on the dose of histotripsy applied. This provides important insights into how histotripsy may enhance immune responses by promoting tumor antigen release, thereby supporting its potential role in cancer immunotherapy. I have few suggestions for the improvement.
- How stable is HER2 expression in the E0771E2 cells over time, and how consistent is it in vivo in the C57BL/6 HER2 transgenic mice? Did you observe any variation in expression levels, and if so, how did you account for this in your analysis?
Response: The HER2 positive cells are cells stably transfected with HER2 (Ref 15). The stable transfection leads to permanent genetic changes that can usually be passed on to future cell progeny (Ref below 1). The addition of G418 can enhance the expression of HER2. Therefore, the expression of HER2 in in vitro E0771E2 cells is stable, leading to the high expression of HER2 in the C57BL/6 HER2 transgenic mice (Ref 15).
- Colosimo A, Goncz KK, Holmes AR, Kunzelmann K, Novelli G, Malone RW, Bennett MJ, Gruenert DC. Transfer and expression of foreign genes in mammalian cells. Biotechniques. 2000 Aug;29(2):314-8, 320-2, 324 passim. doi: 10.2144/00292rv01. PMID: 10948433.
2) How were the different pulse repetition dosages of histotripsy selected? Were these doses chosen based on preliminary experiments, and how do they compare to clinically relevant parameters?
Response: The different tested dosages were chosen because the covered a range of doses used in previous in vivo animal studies (Ref 9,10,13, 17). These doses are theoretically relevant to the treatment in clinical settings but they are not the same as those used in the clinic since the histotripsy system used has a smaller focus and shallow focal distance that is more suitable for the in vitro and in vivo mouse experiments.
3) How does tumor heterogeneity affect the release of HER2 after histotripsy? Were any studies performed to determine if different regions of the tumor had different responses to histotripsy?
Response: No, we haven’t taken potential tumor heterogeneity into question for Her2 release. In the past preclinical in vivo studies and the current clinical treatment, one dose has been used to treat the entire tumor to achieve tumor regression. Prior studies show that the histotripsy dose needed to fractionate the target tissues depends on tissue mechanical properties (Ref 9, 13). Therefore, it is possible that different regions of the tumor with varying properties responds differently to histotripsy. We thank the reviewer for the comment and plan to study them in the future in vivo experiments.
4) Is there any direct evidence from this study that histotripsy-induced HER2 release leads to MHC class I or II presentation by APCs?
Response: Currently there is no direct evidence demonstrating HER2 release leading to MHC class I or II presentation by APCs, which should be one of our future directions. Thanks for the good point.
5) How does histotripsy impact other components of the tumor microenvironment, such as immune suppressive cells or stromal components? Were these effects considered in relation to the release of HER2 and potential immunostimulation?
Response: Unfortunately, in this study we have not yet examined the impact of histotripsy on other features of tumor microenvironment (TME) and this should be another area in our future study. These effects could not be studied in the in vitro and ex vivo experiments. We thank the reviewer for the comment and plan to study them in the future in vivo experiments.
6) The introduction mentions abscopal effects, but does this study provide direct evidence of these effects in the animal model used? If not, how might future studies be designed to assess these effects systematically?
Response: The result of this study does not show the direct evidence to link the release of HER2 and the abscopal effects observed in our early animal models tested. But it is known that the abscopal effects depend on the release of tumor-associated antigens (TAAs) and/or tumor-specific antigens (TSAs)/neoantigens from tumors (below Ref 1-3). Therefore, the histotripsy-mediated release of HER2 from breast cancer cells as demonstrated in this study does suggest its potential role in the abscopal effect that has been observed in animal models treated with the histotripsy (Ref 11,13). There has been both preclinical and clinical evidence of abscopal effect induced by histotripsy (Ref 9, 18). Studies have been planned in our labs for the mechanism of the abscopal affect generated by the histotripsy by a series of experiments such as measuring the in vivo cellular/humoral response to different lysate fractions generated by histotripsy, determining relevant TAAs/TSAs in lysate fractions and sera, co-culturing PBMCs/lymphocytes with the lysate fractions, and observing untreated lung metastasis tumor growth after primary tumor treated by histotripsy.
- Franzese O, Torino F, Giannetti E, Cioccoloni G, Aquino A, Faraoni I, Fuggetta MP, De Vecchis L, Giuliani A, Kaina B, Bonmassar E. Abscopal Effect and Drug-Induced Xenogenization: A Strategic Alliance in Cancer Treatment? Int J Mol Sci. 2021 Oct 1;22(19):10672. doi: 10.3390/ijms221910672. PMID: 34639014; PMCID: PMC8509363.
- Xia QH, Lu CT, Tong MQ, Yue M, Chen R, Zhuge DL, Yao Q, Xu HL, Zhao YZ. Ganoderma Lucidum Polysaccharides Enhance the Abscopal Effect of Photothermal Therapy in Hepatoma-Bearing Mice Through Immunomodulatory, Anti-Proliferative, Pro-Apoptotic and Anti-Angiogenic. Front Pharmacol. 2021 Jul 6;12:648708. doi: 10.3389/fphar.2021.648708. PMID: 34295244; PMCID: PMC8290260.
- Chen M, Quan G, Wen T, Yang P, Qin W, Mai H, Sun Y, Lu C, Pan X, Wu C. Cold to Hot: Binary Cooperative Microneedle Array-Amplified Photoimmunotherapy for Eliciting Antitumor Immunity and the Abscopal Effect. ACS Appl Mater Interfaces. 2020 Jul 22;12(29):32259-32269. doi: 10.1021/acsami.0c05090. Epub 2020 May 22. PMID: 32406239.
Some of the above information have been added into the revised manuscript (the 1st Para, Page 14).
7) The manuscript suggests that released tumor antigens lead to APC presentation and subsequent T cell activation. Any functional assays (e.g., mixed lymphocyte reaction) to assess whether T cells were activated by these released antigens?
Response: No studies have been conducted showing that released antigen can directly stimulate APCs and T cells, but that is one of the next experiments being conducted for this study. There has been prior published studies on T-cell activiation following histotripsy tumor treatment (Ref 9,10).
8) What are some of the limitations of using the E0771E2 mouse model for studying HER2-positive cancer in the context of immunotherapy? How might these limitations affect the interpretation of your findings, and what future work do you propose to address these limitations?
Response: The EO771 cell line is considered a luminal-B breast cancer line that has been shown to have an immunosuppressive TME that may be unresponsive to immunotherapy. Our EO771E2 expressing HER2 may be more immunogenic (Ref 15), however future studies aimed at combination treatment should take this into consideration when choosing potential therapy to combine with histotripsy.
The HER2 transgenic (E0771E2) C57BL/6 mice have a healthy and normal life span without spontaneous tumor growth. Therefore, unlike nude mouse models, the macro- and micro-environment of tumor growth in this established mouse model remains in a natural and normal condition. These features create an ideal experimental situation in which HER2+ tumor cells will naturally grow and can be biologically targeted in a physiologically relevant way that mimics the targeting of HER2+ tumor cells in patients (Ref 15).
The limitation of this mouse model in in the context of immunotherapy is that the regulatory system may be different from human (Ref below 1). The big animal models such pigs can be more relevant to human.
- Bosenberg M, Liu ET, Yu CI, Palucka K. Mouse models for immuno-oncology. Trends Cancer. 2023 Jul;9(7):578-590. doi: 10.1016/j.trecan.2023.03.009. Epub 2023 Apr 20. PMID: 37087398.
9) Were any experiments conducted to determine whether histotripsy selectively targets tumor cells without damaging surrounding normal tissue? This could provide insight into the safety of histotripsy in a clinical context.
Response: No experiments were performed in this study to address this question. However, previous studies (Ref below 1,2) in the field have demonstrated that histotripsy has a sub millimeter targeting accuracy that allows for the generation of precise lesions in numerous animal pre-clinical models. This safety has also been demonstrated in the multi-center clinical trial (Ref below 1,2).
- Vidal-Jove J, Serres X, Vlaisavljevich E, Cannata J, Duryea A, Miller R, Merino X, Velat M, Kam Y, Bolduan R, Amaral J, Hall T, Xu Z, Lee FT Jr, Ziemlewicz TJ. First-in-man histotripsy of hepatic tumors: the THERESA trial, a feasibility study. Int J Hyperthermia. 2022;39(1):1115-1123. doi: 10.1080/02656736.2022.2112309. PMID: 36002243.
- Mendiratta-Lala M, Wiggermann P, Pech M, Serres-Creixams X, White SB, Davis C, Ahmed O, Parikh ND, Planert M, Thormann M, Xu Z, Collins Z, Narayanan G, Torzilli G, Cho C, Littler P, Wah TM, Solbiati L, Ziemlewicz TJ. The #HOPE4LIVER Single-Arm Pivotal Trial for Histotripsy of Primary and Metastatic Liver Tumors. Radiology. 2024;312(3):e233051. doi: 10.1148/radiol.233051. PubMed PMID: 39225612.
10) The study mentions that G418 was used at different concentrations, but higher concentrations caused significant cytotoxicity. Could you elaborate on how the chosen concentration (0.8 mg/mL) balances HER2 enhancement without excessive cell death? How reproducible is this effect across different batches of cells? How might the observed levels of released HER2 affect its immunogenicity in the tumor microenvironment? Was there any attempt to correlate the amount of HER2 released with potential immune activation?
Response: According to a consecutive dose test, the concentration of G418 at 0.8 mg/mL enhance HER2 expression without excessive cell death. The chosen concentration has been reported in literatures (below Ref 1, 2), and our data showed that the expression of HER2 was quite consistent, without any obvious cell death. It is an idea model system for further evaluating the impact of the released HER2 on the immunogenicity in TME with potential immune activation. We will investigate this effect and correlate it with the level of the released HER2 by histotripsy in the future study.
- Yamamoto N, Yang M, Jiang P, Xu M, Yamauchi K, Tsuchiya H, Tomita K, Moossa AR, Hoffman RM. Color coding cancer cells with fluorescent proteins to visualize in vivo cellular interaction in metastatic colonies. Anticancer Res. 2004 Nov-Dec;24(6):4067-72. PMID: 15736453.
- Hoffman, R., Yang, M. Whole-body imaging with fluorescent proteins. Nat Protoc1, 2006,1429–1438.
11) The study shows higher HER2 levels in the cell-free fraction compared to RIPA lysate. Could you elaborate on the mechanistic implications of this finding? Does histotripsy preferentially release HER2 into the extracellular environment rather than leaving it within the remaining cell lysate?
Response: Our interest in the cell free vs RIPA lysed pellet stemmed from a question of whether or not there is an effect of treatment dose on “where” within lysate the HER2 protein is. The cell-free fraction represents HER2 that is ‘cell-free floating’ in the extracellular compartment while the RIPA-lysed pellet represents remaining cells and large cellular debris within the lysate. It would be interesting to know if either or both fractions are necessary for stimulating anti-tumor functions of the immune system.
12) The Western blot results indicate a substantial increase in HER2 release at the high histotripsy dose (~28x increase). Could high-dose histotripsy also induce non-specific protein release or damage to adjacent non-tumor cells? How might this affect the interpretation of HER2 specificity in the tumor microenvironment?
Response: Based on the overall peptide BCA results, we believe that histotripsy is causing the non-specific release of proteins in the extracellular space, not just the release of HER2. While HER2 was the primary target for this study, we believe peptides from other cellular compartments should be detectable in the histotripsy generated lysate.
Round 2
Reviewer 2 Report
Comments and Suggestions for Authors
Since the author did not provide new data nor responded to my comments regarding the experimental design flaws, my review remains the same from last time.
Reviewer 3 Report
Comments and Suggestions for Authors
The authors addressed all my comments. The manuscript is considered suitable for publication.